# Involving Children in Health Literacy Research

**DOI:** 10.3390/children10010023

**Published:** 2022-12-23

**Authors:** Catherine L. Jenkins, Jane Wills, Susie Sykes

**Affiliations:** Institute of Health and Social Care, London South Bank University, London SE1 0AA, UK

**Keywords:** health literacy, health equity, children, child health literacy, Children’s Advisory Group, life course, public involvement, institutional ethnography

## Abstract

Despite the volume and breadth of health literacy research related to children, children’s involvement in that research is rare. Research with children is challenging, but the principles of involvement and engagement underpin all health promotion work, including health literacy. This commentary reflects on the process of setting up a Children’s Advisory Group to consult on an institutional ethnography study of health literacy work from children’s standpoint. The Children’s Advisory Group contributed feedback on the study ethics and design and piloted methods for rapport-building and data collection, including livestreamed draw-and-describe and modified Interview to the Double. Consulting with the Children’s Advisory Group highlighted the importance of listening to children and recognizing and valuing children’s imaginative contributions to methods for involving children in health literacy research. Insights from this commentary can be used to foreground equity-focused approaches to future research and practice with children in the field of health literacy.

## 1. Introduction

Health literacy, as a context-specific social practice [1], has been variously defined [2,3]. Children’s health literacy has been defined in its own right, including what it means for children [4,5]. Children’s health literacy is now increasingly understood as distinct from proximal adult and adolescent health literacy [6], with dimensions specific to the structures and relationships of children’s social location and cognitive and social maturation [7]. The middle childhood stage that follows early childhood and precedes adolescence is a foundational period for independent decision-making and the formation of health attitudes, beliefs and behaviors contributing to health literacy [7,8,9]. 

This commentary understands children’s health literacy as a social practice [1,10] spanning the middle childhood developmental stage [7,11] and functional, interactive, and critical levels [12]. It recognizes the differential demographic patterns, inequalities, epidemiology and health perspectives and dependency within power structures [13] that influence the opportunities available for children to develop health literacy ethically and in children’s best interests [14,15]. It also discusses the neglected domain of children’s critical health literacy [16,17,18,19].

Studying children’s health literacy and how it can be developed is important, because children for whom health promotion messages are too far removed from their contextualized understandings of health may be more likely to struggle to apply and benefit from health literacy in their everyday lives [20]. Children’s everyday lives place them in situations requiring critical decision-making about health spontaneously, ‘in-the-moment in a hallway or after school on a playground, or alone without adult guidance’ [21] (p. e194). When outsourcing their information needs to adults is not feasible, or is discouraged [22], children apply their embodied knowledge to challenge health information irreconcilable with their lived experience [20]. Consolidating this knowledge early in the life course is of benefit to the everyday health-related work that many children are already doing [23], or may need to do in future [24]. However, there is a lack of empirical research on how health literacy can be developed in children that involves children in the research process. 

This commentary starts from the provocation ‘where is the child in child health literacy research?’ [11], and its purpose is to provide practical strategies for actively centering the child in children’s health literacy research. It reflects on the process of setting up a Children’s Advisory Group to consult on the ethics and design of a doctoral study. The doctoral study explored the potential for public libraries to be supportive environments for children’s critical health literacy development, and the work involved in this [25]. 

The methodology for the study on which this commentary is based was institutional ethnography, an equity-focused framework for inquiry into the social organization of people’s work. Institutional ethnography (IE) draws on the standpoint of people in a particular social location to analyse how the knowledge available to them for their everyday work is socially organized in ways that extend beyond their purview. In IE, ‘work’ encompasses any purposeful activity that requires resources, time, and effort to get done [26]. The IE for the study starts from children’s standpoint, and therefore from their knowledge. It recognises children’s activities as ‘workful’ [27], serves as a reminder to keep the research anchored in the interests of that standpoint group, and takes an interests-based approach [14,28] that prioritises children’s ‘bodily experience, relevancies, and everyday knowledge’ [29] (p. 2). IE is aligned with key features of a child health equity implementation framework, including in-depth inquiry into the social organization of children’s contexts and relevant factors of children’s experiences [24]. 

The lack of IE research that foregrounds children’s interests is concerning, especially in the context of a corresponding lack of ‘health literacy research *with* children’ [30] (p. 594, emphasis in original). Learning from children’s standpoint in an IE-informed approach to health literacy research therefore ‘expands the range of interlocutors’ [31] and makes space for children’s voices to enter into the health literacy research conversation [32,33,34]. The IE concept of standpoint facilitates ‘a child perspective’ on the part of the researcher, who learns from children’s experiences and is led by children’s interests [35]. 

Children’s standpoint is under-represented in IE research, despite the inclusion of children in illustrative examples early in the development of IE as a research framework [26]. There are more IEs ‘about’ children [36,37,38,39,40,41] than there are from children’s standpoint, and IE studies that adopt the standpoint of children tend to focus on older children (adolescents) [42,43] or very young children [44,45], rather than middle childhood. Of the few examples that adopt standpoints from middle childhood, two focus on homework practices in families: one prioritizes eight-year-old children’s rights to decline to participate in research [46], and the other explores the work of children aged ten to 16 in shared custody arrangements, including managing homework across different households [47]. 

Strategies for eliciting the work children are involved in, or refrain from, are nascent in IE [48]. Making available multiple options for how children can choose to provide information about the work they do in the everyday settings where they spend time increases researcher workload, but is necessary for learning the details of children’s health-related work in-depth and the social organization of their work and the work of others. 

Patient and Public Involvement and Engagement (PPIE) refers to research carried out ‘with’ or ‘by’ members of the public, rather than ‘to’, ‘about’ or ‘for’ them [49]. The lack of children’s perspectives on the development of research into children’s health literacy makes it imperative that children have a role in subsequent research to ensure that research reflects the lived realities of children’s lives [23]. Best practice guidelines for integrating PPIE with children in UK health research are available [50] and have been reviewed and updated with input from Children’s Advisory Groups. 

Involving a Children’s Advisory Group (CAG) in the design of health literacy research offers a route through which children can advise on the research and see their advisory work being taken seriously and making a difference to how the research proceeds [51]. However, CAG involvement in research related to children’s health literacy is under-utilized. IE’s use of standpoint is an inadequate substitution for PPIE, and although IE has been used to analyse PPIE [52], examples of PPIE with children as part of an IE are limited. Prioritising the involvement of children is needed to advance PPIE in IE, and in studies of children’s health literacy. 

## 2. Consulting a Children’s Advisory Group to Involve and Engage Children in Health Literacy Research

### 2.1. Ethics, Recruitment, and Structure of Consultations

Ethics approval for the study on which this commentary is based was secured from London South Bank University Ethics Committee (ETH2021-0003). Findings from the study are published separately [25]; this article focuses on the processes involved in setting up the CAG and how consulting with the CAG informed how the study was conducted. All child participants and their adults gave permission for children’s contributions to be reproduced. In the absence of an IE-specific guideline, consulting with a CAG for the study was informed by PPIE literature [28,53,54,55,56,57] and grey literature on conducting ethical and inclusive online research with children using Zoom [58,59].

The degree of children’s involvement in the CAG was deliberately aligned with the ‘Consult’ level of the modified International Association for Public Participation (IAP2) spectrum, which requires the researcher to commit to ‘keep you informed, listen to and acknowledge your concerns and aspirations and provide feedback on how your input influenced the research’ [60]. Consultation entails eliciting children’s views to inform decision-making [61,62] and sits on a continuum that includes collaboration (e.g., participatory action research) and child-led shared decision-making (e.g., children as co-investigators/peer-researchers). The use of consultation in designing the study was pragmatic, to enable meaningful PPIE with children to the extent manageable under the constraints of COVID-19 while maintaining researcher control over the timescale of the project. The CAG was put in place to consult with children on the design of a proposed IE and to facilitate the recruitment of children to the final study. The intention in forming the CAG was to support and learn from children as capable and active practitioners of health literacy [16], as well as to build children’s research literacy [63]. 

The CAG for the doctoral study comprised eight children aged seven to 11 years old (middle childhood) [6]. Each CAG member held the job title of Child Advisor (CA). Recruitment of CAs was through referrals by adults who knew children and showed social media posts about the study to them. PPIE with children at the Consult level in this multiphase study was not a one-off occurrence [61]: four consultations between 20–60 min were held with each CA or small-group twos or threes online during March–May 2021, followed by a debrief in November 2021. The CAG did not meet collectively, in order that each child or sibling small-group had time and space to make their substantive contribution. 

The online nature of the CAG allowed the research to progress while social distancing measures were in place. Zoom was selected as a teleconferencing app that children were already familiar with from its use in homeschooling during COVID-19. The option to consult over telephone was also offered, to mitigate digital exclusion. Consultations were audio-recorded and transcribed in real time using Otter.ai transcription software, with live captioning visible on Zoom. Parents/caregivers of CAs were welcome to join the call, and CAs and their adults both provided informed consent prior to each consultation.

CAs chose their own pseudonyms (explained by the researcher to the CAG as “research codenames”). The pseudonyms reflected children’s individual passions, e.g., outer space (White Hole) and YouTube influencers (KSI), and children expressed the intention to search for their codenames in open access outputs from the study. Table 1 summarizes the composition of the CAG.

The first consultation was used for introductions. Prospective CAs could find out more about getting involved in the CAG, inform the researcher of their pseudonym, and practise signing in to Zoom using the pseudonym as their virtual name-badge. The CAs provided feedback on the potential worth of the study for other children of the same age (i.e., that the study’s focus on critical health literacy development in children was deemed important and relevant) [64], and indicated their preference to receive cashless incentives and to be kept up-to-date in between consultations by post to reduce screen-time. The scope of the CAG was discussed to manage CAs’ expectations of what the CAG would be able to achieve within the timescale of the project, and to clarify their responsibilities and job descriptions as consultants. The concept of catalytic validity [64] guided the provision of opportunities for CAs to critically reflect on the determinants that constitute their own and others’ health chances [65] and analyze possibilities for change without subsequent ‘action paralysis’ [66] in face of the difficulties in feasibly implementing such change from their social location. 

The second consultation with CAs focused on ethics, specifically CAs’ confidence in the appropriateness and safety of the research (framed as, ‘would CAs be happy for their siblings/friends to participate in the later study?’), and edits to the recruitment and consent documentation.

The third consultation involved CAs in piloting two research tools: a rapport-building activity using draw-and-describe, and a data collection method using a modified Interview to the Double technique. 

### 2.2. Listening to and Applying Children’s Methodological Contributions

Draw-and-describe has previously been used in health research with children [67] and lends itself to producing insights into understandings of complex or multidimensional concepts (such as critical health literacy) [68]. While this method has been challenged [69,70] and should ideally be used alongside alternative ways for children to engage, it was useful for familiarizing CAs with talking about their understandings of critical health literacy and for learning from them how best to introduce critical health literacy as the research topic to children in the later study. 

Interview to the Double (ITTD) combines in-depth interviewing (‘tell me what you do’, ‘walk me through a day-in-your-life’) with observation (where the researcher traces what the informant is observed as doing in practice onto what the informant has told them about what they do). It seeks to learn what work the informant does day-to-day, and how they know what to do in the first place, in sufficient detail that the researcher could replace them in their daily routine the next day (as a body-double or doppelgänger—the ‘Double’ of the technique’s name). ITTD can ‘reveal, question, challenge and offer perspectives that run counter to what we think we know’ [33] (p. 6). It also renders accessible ‘a child’s perspective’: the child’s views on the experiences they identify as relevant (differentiated from, but supplementing, ‘a child perspective’, or the child standpoint as adopted by adults) [35].

ITTD is informed by practice theory [71] and has been used in studies of information literacy practices in the library and information science (LIS) field [72]. A method similar to ITTD appears in early fieldwork involving IE’s founder [73], where it was used to—understand the everyday practices of specialized workers without resorting to jargon or imprecise language that would obscure or displace those practices. It has also been used as part of work-based interviews in IE [74]. ITTD’s integrated observation component makes it a pragmatic option during a pandemic, when opportunities for observation are limited. While there is some precedent for using ITTD with children [75], and its use can help redress the power imbalance in an interview situation where the child is confronted with an unfamiliar adult [63], children are not routinely involved in designing interviews for health literacy research.

The final consultation focused on child-accessible public engagement strategies for disseminating the study’s findings and included a formal debrief. All CAs received a personalized certificate thanking them for their work (that could be included in their school records of achievement, as requested in the initial consultation), and an incentive pack: a reusable tote bag containing materials that children could use to conduct their own health research projects. 

Evaluation forms were provided to CAs following each consultation so that the researcher could take steps to rectify any issues ahead of the next consultation. The evaluation form template was based on the Lundy Model of Participation [76], updated to include items for evaluating children’s experiences of remote research [77]. The forms served to hold the researcher accountable by creating conditions under which it was unacceptable for the researcher to elicit children’s views and then ignore them. Figure 1 shows an evaluation form completed by a CA. 

Data gathered in consultations included the transcripts, children’s text in Zoom chat, drawings, and evaluation forms, and a reflective log filled in by the researcher immediately after each consultation and during the transcription process. Following the final consultation session, CAs were debriefed and their contributions were reviewed alongside the reflective log in order to identify insights.

## 3. Insights from Consulting a Children’s Advisory Group 

The lessons learnt from consulting with a distributed CAG are reported in this section with supporting extracts and media from the recorded consultations. Three insights are identified: the role of adults when researching health literacy with children requires a balance between providing assistance and learning from children’s contributions; children’s preferred involvement in research is being part of a collective or group; and involving children can help develop research methods that are child-friendly. 

### 3.1. Safeguarding Children’s Involvement and Engagement in Health Literacy Research

There are numerous ethical considerations when conducting health literacy research with children, including the frequent use of schools as recruitment grounds and settings for research activities; in the school setting, children may find it difficult to decline to take part. Consultations were scheduled during school closures in the first wave of the COVID-19 pandemic in the UK, so schools’ and teachers’ influence was minimal. However, the role of accompanying adults (parents/caregivers) as gatekeepers and “translators” for children significantly shaped CAs’ involvement.

The importance of process consent with children was emphasized in one CA’s response to their previously signed consent form being screen-shared by the researcher for review:

I’m pretty sure my mum did that. I don’t remember doing it.(Jar Jar Binks)

CAs perceived the consent form as a barrier to the involvement of other children in the study. CAs questioned the five-year data retention stipulated on the consent form, and highlighted problems in the informed consent process as experienced by CAs themselves. The consent form template provided by the University Ethics Committee was extensively revised in response to CAs’ feedback. These revisions enhanced the health literacy claims of the research by modelling health-literate practices in information provision and included the replacement of legal jargon and boilerplate text with edits made by CAs verbally during the Zoom sessions, or asynchronously after the session. 

Figure 2 shows marked-up drafts of the consent sheet and part of the participant information sheet for the study, to which a CA lent their critical eye. The CA’s comments were transcribed by the CA’s accompanying adult. 

Edits included the use of ‘happy’, ‘confused/unsure’ and ‘not happy’ emoji against each consent clause and a ‘thought bubble’ space on the form where children could expand on their reasons for selecting each emoji, or jot down any questions. However, these edits were transcribed by an adult, and were therefore presented at a remove from the child’s original feedback.

Parental/caregiver involvement required careful management. On the one hand, adults were prone to fill silences if they felt that their child was “pondering” (taking too long to answer), or to interrupt to keep the research conversation “on track”: 

Catherine said any questions!(White Hole, to parent)

You’re going to have your work cut out here.(Parent, to Catherine [the researcher])

On the other hand, parental/caregiver input into the CAG was essential for facilitating the return of consent documentation, providing correct postal address details, managing in-call mishaps (like a dropped ice-lolly), amplifying comments from CAs whispered in their ear, suggesting pre- and post-consultation reflections, and suggesting alternative ways for children to provide their opinions:

For homeschooling, they’ve not been allowed to type stuff in chat […] [to Tigerlilly] you might type opinions in chat mightn’t you rather than saying them?(Parent)

[Tigerlilly whispers in Parent’s ear]

Yeah, so it might be, so it’d be good if Catherine can make this feel like not school.(Parent)

The word ‘critical’, as used in the study’s focus on critical health literacy, required parent/caregiver input before children were able to contribute examples of critical health literacy meaningful to them: 

Wait, was does critical mean again?(White Hole)

[…]

What’s the word again? It’s the word that that tells you questioning if it’s real, or not.(White Hole)

Okay, so she [Tigerlilly] thinks it sounds like it will be really useful, but they still don’t quite absolutely understand what it will, what that what it would look like.(Parent)

the critical bit.(Tigerlilly)

Children’s engagement in health literacy research is contingent on adults’ support. The presence of parents/caregivers both hindered and helped consultation with children.

### 3.2. Consulting as Part of a Collective Is Valued by Children

The ‘G’ for ‘Group’ in CAG is important. CAs communicated that although the one-to-one or small-group consultations enabled them to share their views, they missed the collective aspect of being part of a group:

Only thing I would say is I wish we could be there together.(ASDPENGUIN22)

I’d like to see the other people.(Jar Jar Binks)

The balance between enabling each CA sufficient space to speak (as in 1-to-1, pair, or small-group consultations), and the interplay of ideas in a larger group, requires careful consideration. While individual and small-group sessions enabled each CA to be heard, CAs expressed disappointment at not sharing their journey with other CAs and the researcher missed observing how convening the whole CAG together might have encouraged CAs to interact with each other or pool their combined knowledge. 

Crediting children’s contributions as a group was a topic of discussion in the CAG sessions. CAs were interested in tangible evidence of how their contributions would be recognized in the research: 

Include like what we’ve thought of, stuff we’ve come up with.(KSI)

Children valued formal acknowledgement of their collective work of, as Tigerlilly described it in her edits to the recruitment documentation, ‘Helping Catherine from a Child’s Perspective’. 

### 3.3. Methods That Work for Facilitating Children’s Participation in the Health Literacy Research Process 

Consulting with children led to the development of two imaginative research methods that appealed to children: livestreamed draw-and-describe, and Interview to the Alien. 

The draw-and-describe exercise aimed to facilitate rapport-building and was piloted with the CAG as a two-part activity using a blank cartoon strip comprising three boxes. The first part of the activity asked CAs, in advance of the consultation, to draw a child with a mask (see Figure 3 for a completed Drawing 1). A COVID mask was not specified in the instruction, but all CAs chose to draw a COVID mask, as confirmed in their accompanying commentaries. In the consultation session, CAs were invited to complete the cartoon strip. Drawing 2 involved the children in creating an alien cartoon character who wanted to know why the child was wearing a mask for their and others’ health. Drawing 3 located the alien in a public library setting (the case selected for the wider study) and asked CAs to draw the child showing the alien how to navigate this setting for ‘Earthling’ and alien health, including, e.g., how to ask library staff for help to display an awareness-raising poster, identify misinformation, or contact a local politician about masks contributing to litter problems locally. 

Consultations with the CAG linked this method to CAs’ stated expertise and interest in livestreaming via video-based social media platforms like YouTube Kids or TikTok for Younger Users. Piloting this method with the CAG led to changes to the phrasing of the draw-and-describe instructions [78] to reframe it in terms familiar to children, such as ‘an Instagram Story/Reel of your day’, ‘a livestream on YouTube’, or ‘a TikTok how-to video’ (using the versions of these platforms for younger users). CAs suggested the instruction ‘Pretend you’re doing a livestream for [child’s preferred platform] with a running commentary of what you’re drawing. Position your camera to show the drawing taking shape’ as a prompt for children to, e.g., explain their decision to make edits by erasing part of their drawing, or flag what ASDPENGUIN22 called a “spoiler alert”, and Tigerlilly ‘the big reveal’, as their drawing developed. Framing the draw-and-describe activity in livestreaming terms helped to maintain a focus on children’s embodied reflections in an online environment, and the drawing component of the study was a self-reported determining factor in the decision by children to participate in the later study.

The idea to incorporate the alien as an interlocutor in the draw-and-describe activity was suggested by CAs, who welcomed the “randomness” of the alien and thought it would appeal to other children: ‘It’s just completely random I love it’ (Jar Jar Binks). Aliens (‘Mork’ and ‘Og’) have previously been used as stimuli in a health literacy intervention aimed at children [20]. The CAs separately agreed that the alien here should be children’s own creation: 

cos then you can design it like, if they [adult researchers] designed it might be something like, it likes reading books. But you wanted yours to be, like, not doing that.(Tigerlilly)

The popularity of the alien in the livestreamed drawing exercise led to its being retained as a proxy for the researcher in the data collection tool. Consultations around the data collection tool developed from the researcher pitching to CAs a semi-structured interview technique: ITTD. 

The CAG members, taking the ‘Double’ of the ITTD literally, were understandably skeptical about the researcher’s ability to plausibly “get away with” replacing them in their daily routines. Better by far, CAs suggested, would be to ask children to educate the researcher as if they were educating an alien who knew nothing about how daily life works on planet Earth:

It’s like explaining to someone that’s not educated. And I think that if there’s an alien in there, it makes the story more interesting.(ASDPENGUIN22)

The ‘Double’ was accordingly switched to the ‘Alien’ in a modified version of the ITTD to create Interview to the Alien (ITTA). ITTA situates children as authoritative knowers in contrast to the alien, who knows very little and is reliant on children sharing their experiences. Its creation is indebted to the CAs’ imaginative contributions.

### 3.4. Challenges in Equity-Focused Research with Children 

Challenges in recruiting children to the CAG included its online nature. Despite efforts to involve children without access to a WiFi-enabled device by using alternative means of communication, such as telephone calls or by post, some children were still excluded (e.g., children from the Liveaboard Boater community without telephone credit or a fixed address). Challenges like this demonstrated the importance of referring to an equity-focused framework [24] when planning to convene a CAG, as well as the importance of ensuring that health literacy research with children does not inadvertently reproduce inequalities [79] and takes intersectionality into account in recruitment strategies. It is therefore key to offer prospective CAG members more than one option to participate in the research, and to integrate space for reflection and feedback within consultations that do not rely on evaluation forms (Figure 1). Taking an equity-focused approach is particularly relevant for research with children that is focused on critical health literacy [79].

## 4. Discussion

Changes made through consultation with the CAG helped ensure that the proposed research would be ethical (i.e., CAs would be happy for their siblings/friends to take part) and relevant to other children in the same age-range. Changes included verbal and written edits to the documentation used to recruit and consent child standpoint informants and help them understand their rights in the research process; ideas for disseminating the research so that other children would see it (a slide-deck for school assemblies and a YouTube Kids video); and formulation and refinement of tools for introducing children to the research topic of critical health literacy (a draw-and-describe exercise) and for use in data collection (ITTA, a modified combined interviewing and observation technique inspired by ITTD). The CAG also contributed to methodological development in IE from children’s standpoint.

There are some precedents for children’s involvement in health literacy research, including the involvement of children who are unwell [80], children in good health [20], and Young Carer Health Champions [81] (Young Carers are children aged under 18 who provide unpaid care to another person of any age). During the initial waves of COVID-19, an international research collaboration used drawing elicitation as a rapid research method to understand the information available to children about the pandemic. The drawings collected from children in England depicted children’s actions as protecting themselves, their families and wider society [82]. The study linked to this commentary also elicited drawings that captured children’s critical health literacy knowledge [25] and involved children in work-based interviews that recognized children’s work: as research advisors, and as health literacy practitioners.

However, PPIE work with children in health literacy research remains rare. Measures frequently do not include PPIE input beyond testing of instruments, e.g., the cross-national Health Behaviour in School-Aged Children (HBSC) survey (which collects data on the health and wellbeing, social environments and health behaviors of children aged 11 and over, and in which the Health Literacy for School-Aged Children–HLSAC instrument is an optional supplement) [83]. Dyadic studies conducted and published by adults to meet adult-led professional development objectives and research norms have tended to use the health literacy of proximal adults (e.g., parents/caregivers and teachers) as a proxy for children’s own [84]. 

Guidance for translating consultations with children into outputs that can make a wider difference and attract the attention of the health literacy field is also scarce, as the available routes–such as co-authorship on a published paper–are not set up to facilitate children’s involvement in them. While there are precedents for involving children as co-authors [56], it remains the case that publishing workflows and metadata fields feed into perpetuating research norms that complicate articulating and evidencing children’s contributions in ways that conform with the contributor roles recognized by standards such as CRediT (Contributor Roles Taxonomy). 

In consequence, there is a lack of nuanced understandings of children’s health literacy, particularly their critical health literacy: how they access and appraise information, how they apply that information in practice, and what is most relevant and important to them. The CAG’s contributions highlight the need for future research practice to address systemic barriers to children’s involvement in health literacy research at every stage of the process, from ethics documentation and recruitment procedures through to dissemination. Consulting with the CAG has also demonstrated the value of involving children as advisors on how children’s standpoint can be sought and understood in IE through methodological innovation with members of that standpoint group. 

While the logistics of facilitating children’s involvement in research mean ‘there is always an adult present *somewhere*’ [34] (p. 6), children’s unfiltered contributions should be supported and acknowledged so that the health literacy field, and the ongoing development of IE, can continue to learn from their insights. 

## 5. Limitations

The timing of the study that forms the focus of this commentary (during COVID-19) meant some CAs having more availability and resources than others to participate in consultations online. However, the CAs who were able to join consultation sessions represented significant information power [85], meaning that the information available from their first-hand and diverse experiences somewhat mitigated the small sample size [85,86,87] that also kept postal communications with the CAG manageable. Distributed CAGs should be consulted on whether to include plenary sessions that give children the opportunity to meet the others working alongside them. Furthermore, critical health literacy is a relational practice that can be enhanced by being conducted in a group, as findings from the wider study to which the CAG contributed have also concluded [25].

Demographic data were collected, but did not extend to a formal question on whether or not a CA had participated in health-related research before. In light of the inverse information law, this information would have been useful to guide future recruitment priorities for CAGs in health literacy research. 

## 6. Conclusions

It is important to redress children’s lack of involvement in health literacy research. Convening an online CAG, where children are involved in research at the consultation level, can contribute to this if the CAG meets as a collective and the role of accompanying adults is carefully managed so that the researcher can learn directly from children. Methods to engage children and support them in sharing their views are best developed in consultation with children themselves. Taking an equity-focused approach that reduces barriers to participation and values children’s information power and experiences has implications for future ways of working, such as normalizing early CAG involvement in health literacy research proposals and drawing on children’s standpoint to enrich adults’ knowledge and understandings of children’s health literacy. 

## Figures and Tables

**Figure 1 children-10-00023-f001:**
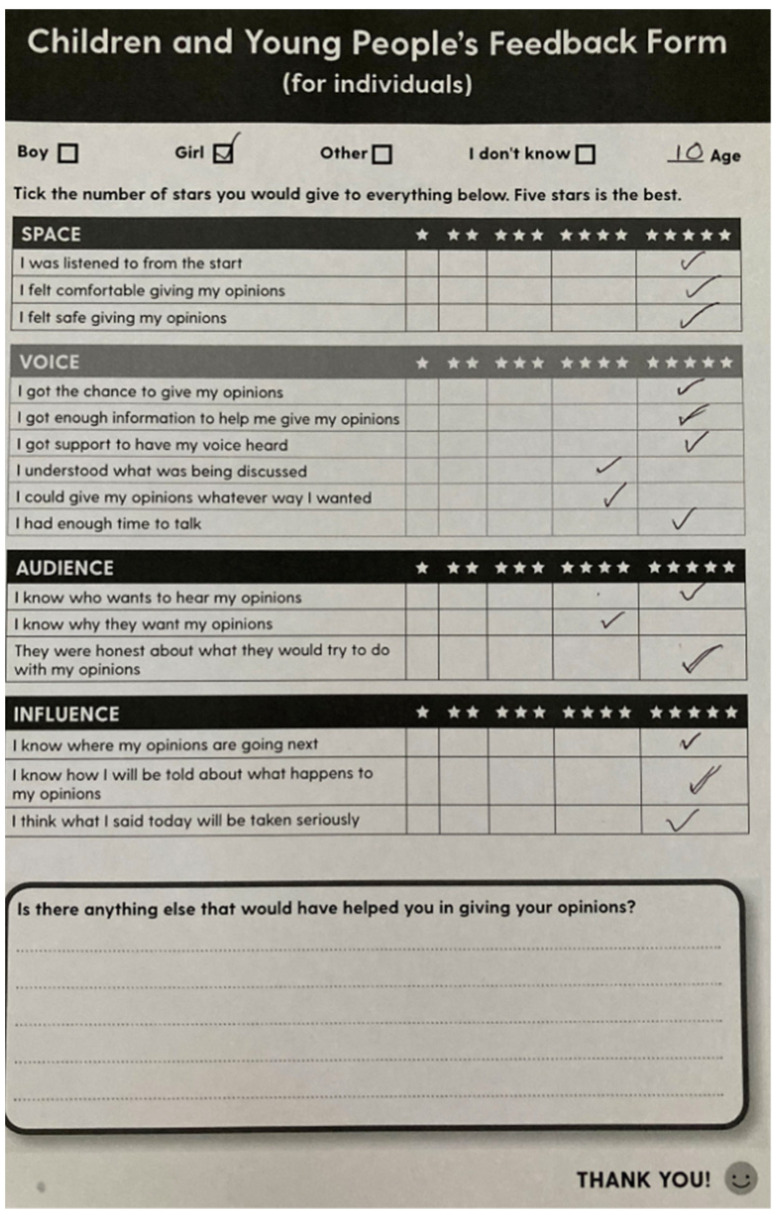
Evaluation form completed by a Child Advisor.

**Figure 2 children-10-00023-f002:**
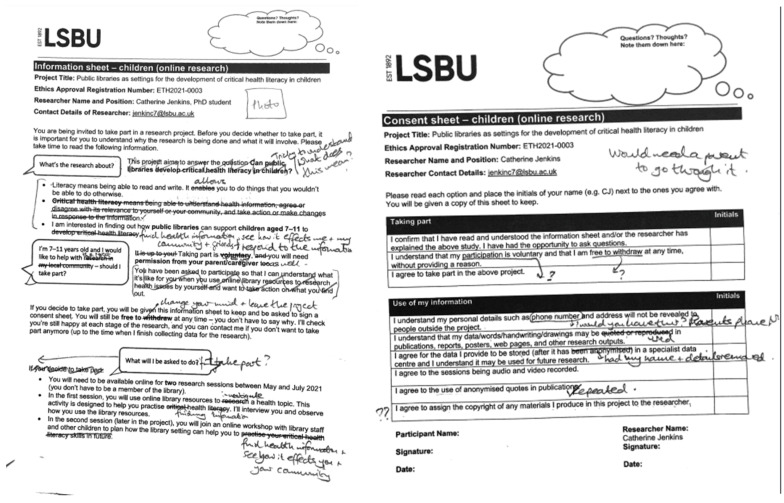
Edits by a Child Advisor dictated to, and transcribed by, their parent/caregiver.

**Figure 3 children-10-00023-f003:**
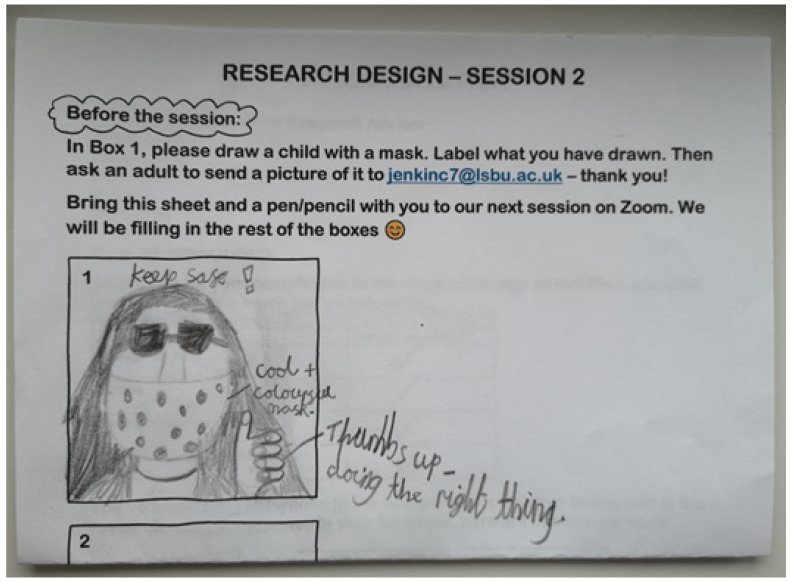
Example of a Child Advisor’s pre-consultation ‘drawing 1′. The labels read: ‘keep safe!/cool + colourful mask’/‘Thumbs up–doing the right thing’.

**Table 1 children-10-00023-t001:** CAG demographics.

Child Advisor Pseudonym	Age	Gender
Luna Starshine	7	M
Jar Jar Binks	8	M
White Hole	8	M
ASDPENGUIN22	9	F
Ronaldo	9	M
KSI	10	M
Tigerlilly	10	F
Willowshot Ebony	11	F

## Data Availability

Not applicable.

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
