# Peer review of "Involving Children in Health Literacy Research"

_children, 2022, doi:10.3390/children10010023_

Round 1
Reviewer 1 Report
This paper adds to the evidence that there are need for children’s involvement in the health literacy research
There are a number of issues with the manuscript which that should be considered:
Abstract
· Please provide further information about the study outcomes
· In the abstract you need to answer the following questions, what, why and how and discuss the study new findings, limitations, and future research
· The abstract should state briefly the purpose of the research, the principal results and major conclusions. An abstract is often presented separately from the article, so it must be able to stand alone
Introduction
- discuss the research aims, research gap and discuss the paper layout Add up-to-date references to support your discussion
- The necessity and innovation of the article should be presented to the introduction
- The literature reviewed and cited is in the main rather old. There are about many recent researches published on this topic, please cite the following articles:
Methods
· The methodology of this study should be detailed, limit information was provided on method and materials.
· How the author could improve the internal and external validity of the study.
· How the Author could generalize the study finding
Discussion
- I believe that more in depth discussion is needed. The discussion as present now is simple and concise. Revision of more papers using similar technique is needed
- In the discussion, please discuss if the study research questions are answered or not Also introduce the model in detail. Draw a conclusion from this study and present the limitations and future research.
- . The major defect of this study is the debate or Argument is not clear stated in the introduction session. Hence, the contribution is weak in this manuscript. I would suggest the author to enhance your theoretical discussion and arrives your debate or argument
- Please make sure your conclusions' section underscore the scientific value added of your paper, and/or the applicability of your findings/results, as indicated previously.
- Please revise your conclusion part into more details. Basically, you should enhance your contributions, limitations, underscore the scientific value added of your paper, and/or the applicability of your findings/results and future study in this session
- Please provide the research implication (what this research add ?)
- Study limitation should be added
Reviewer 2 Report
Some comments:
1. In the Introduction, it is desirable to state the specific purpose of this research.
2. The Introduction lacks a description of the concept of health literacy itself and its research possibilities. How do the authors define children's health literacy? It is indicated that "Children’s health literacy is increasingly recognized as a practice distinct from adult and adolescent health literacy" [lines 20, 21], but it is not indicated how the authors understand this concept within the study. Perhaps these publications can be helpful: Borzekowski, D. (2009). Considering children and health literacy: A Theoretical approach. Pediatrics. 124(3), S282–S288. Bánfai-Csonka H, et al. (2022) Health Literacy in Early Childhood: A Systematic Review of Empirical Studies. Children. 9(8):1131.
3. The study appears to emphasize the explorative approach to the conceptualization of health literacy rather than the theory-driven conceptualization of health literacy (Frisch, A. L., Camerini, L., Diviani, N., Schulz, P. J. (2012). Defining and measuring health literacy: how can we profit from other literacy domains? Health Promotion International. 27(1), 117–26. Jordan, J. E., Buchbinder, R., Osborne, R. H. (2010). Conceptualising health literacy from the patient perspective. Patient Education and Counseling. 79(1), 36–42). Perhaps it is worth mentioning and noting that the study is conducted in the interpretative research paradigm (because the positivist paradigm is mentioned in the discussion [line 387].
4. The introduction could be more focused, as it often repeats the premise that children are not represented in health literacy research. The idea of “child perspective” and “child’s perspective” can be useful in this context (Soderback, M., Coyne, I., Harder, M. (2011). The importance of including both a child perspective and the child's perspective within health care settings to provide truly child-centred care. Journal of Child Health Care. 15, 99–106). In addition, 'child as an agent' (Velardo, S., Drummond, M. (2017). Emphasizing the child in child health literacy research. Journal of Child Health Care. 21(1), 5–13).
5. In general, a better theoretical foundation for health literacy as a social practice1 and child- centred (health) research would be desirable both in the Introduction and in the Discussion.
6. It would be better to divide the method part into subsections (participants, sampling strategy, data collection, data analysis) so that the text is easier to understand.
7. What is the design of this study? Is it qualitative ethnographic research? The samp le of study participants is small, despite being a qualitative study. What sampling method was used? What were the selection criteria? What method(s) of data analysis was used ((preferably with references)? The text written on lines 192-196 seems to be insufficient. What method was used to identify the themes [line 196]? Was it (inductive/deductive/hybrid) thematic analysis, content analysis, framework analysis?
8. In general, reading both the results and the discussion part, one does not leave the feeling that health literacy has been used only as a brand, but the study itself actually describes the participation
1
Papen, U. (2009). Literacy, learning and health – a social practices view of health literacy. Literacy and numeracy studies. 16(2), 19–34.
Bertschi, I., Sahrai, D. (2016). Ethnographic insights into health literacy as social practice in vulnerable families in Switzerland. European Journal of Public Health. 26(1) (abstract)
amerski, S. (2019). Health literacy as a social practice: Social and empirical dimensions of knowl-edge on health and healthcare. Social Science and Medicine, 226, 1–8.
of children in the study as such (not specifically in health literacy research). The context of health literacy is missing or not well presented.
According to the authors, to what extent characteristics of the child's cognitive and social development affect their health literacy abilities (especially interactive and critical health literacy). Also, can there be a situation where children's cognitive and social capabilities are overestimated, creating overload and excessive responsibility in the context of health (literacy) research. This is an old but very good article that might be helpful: Kalnins, I., McQueen, D. V., Backett, K. C., Curtice, L., Currie, C. (1992). Children, empowerment and health promotion: some new directions in research and practice. Health Promotion International. 7(1), 53–59.
Round 2
Reviewer 1 Report
The authors have satisfactorily addressed all my concerns in my previous reviews.
Author Response
Thank you for confirming that we have satisfactorily addressed your concerns. We have also made improvements to the English, as requested.
Reviewer 2 Report
Thanks for the improvements! I hope that they will make the direction and content of the overall work/paper/manuscript more purposefully focused on children's agency and research in health literacy.
The methods part is still a bit difficult to understand, but maybe it is due to not knowing the local context.
In general, I wish to maintain and strengthen this unique perspective of the child's active role in health literacy research to obtain the children's own vision and knowledge, which can exist in parallel with the so-called expert (adult-centred) knowledge of children's health literacy.
Author Response
Thank you for your kind words, your comments were very helpful and we agree they have improved the structure and focus of the paper. We have also aimed to clarify the Methods, as requested.